# Hypergraph modeling of complex interactions: Applications from human musculoskeletal structures to complex system dynamics

**Hiroko Yamano** [1]*, **Shu Liu** [2], **Fujio Toriumi** [2]

**1** Institute of Future Initiatives, The University of Tokyo, Tokyo, Japan, **2** Department of Systems Innovation, School of Engineering, The University of Tokyo, Tokyo, Japan

* yamano@ifi.u-tokyo.ac.jp

**Data Availability Statement:** https://github.com/ryusukemomota/nanatexrepository name: nanatex.

**Funding:** The author(s) received no specific funding for this work.

## Abstract

The musculoskeletal network is a complex system of different types of nodes and edges interacting with each other. Although there is a wealth of knowledge about the anatomical components of the human body and the connections between them, the interdependence of these components as a system remains largely unexplored. This study aims to understand the structure of musculoskeletal networks by using hypergraphs as a model of the musculoskeletal system with many-to-many connections. We used both pairwise and hypergraph-based embedding methods to learn the connectivity of muscles. Experiments demonstrated the superiority of the proposed hypergraph-based method over pairwise methods in distinguishing the specific roles of the muscles connecting different body parts.

## Introduction

The human musculoskeletal system is characterized by interwoven units of heterogeneous body parts. Numerous researchers have studied the composition of the human body, attempting to understand each element that makes up its structure and the mechanisms by which it functions. Indeed, the anatomical understanding of the human body has advanced greatly by separating the components one by one [1]. These anatomical approaches have described muscle function by isolating individual muscles, joints, and their responses to specific injuries on the skeleton. However, despite the great success of reductionism in modern science, the holistic understanding, based on the complex interactions of the elements, of how the multiple elements function interdependently to affect the body system remains largely unknown [2, 3].

The connection between muscle and bone is not inherently one-to-one. Rather, it is a many-to-many interaction as known in the mapping between brain and muscle [4, 5]. For example, the humerus is connected to several antagonist muscles responsible for flexion and extension movements, and the coordinated action of these muscles allows the complex movements of the elbow joint. While the primary muscles responsible for supporting flexion of the elbow are the biceps brachii, brachialis, and brachioradialis, those responsible for extension are the triceps brachii, all of which are connected to the humerus. On the other hand, the

**Competing interests:** The authors have declared that no competing interests exist.

latissimus dorsi (wide back muscles) are connected to several bones, including the spine, ribs, and pelvis, and play an important role in stabilizing the upper body. This multiple connectivity between muscles and bones implies that the usual pairwise representation used in network science is inadequate to describe real situations of higher order interactions that occur in the body system.

In recent years there has been a growing application of network science to elucidate the complexity of the living system [6]. Although limited in number, several studies have examined the body system to enrich anatomical understanding with a variety of network representations [7]. For example, the musculoskeletal system can be represented by a bipartite network [8], where the nodes are either muscles or bones and the edges are the connections between them. In this network, there are two types of nodes (muscles and bones), and the edges between them are not accompanied by physical substances. In order to directly show the relationships between a particular set of nodes, researchers often use one-mode projection, which transforms a bipartite network into a unipartite network [9]. There are two ways to project the connections of the bipartite muscle-bone network. One is a muscle-centric representation, where muscles are the nodes and bones are projected onto the edges according to their adjacency, and the other is a bone-centric projection, where bones are the nodes and muscles are the edges.

Since the one-mode projection always loses some of the information that existed in the original bipartite network, researchers have investigated appropriate weighting methods to reflect the structure of the projection network [9–13]. These studies include weighting methods that directly count the number of common neighbors [10], discount the weights according to the degree of common neighbors [11], discount by both the degrees of nodes and their common neighbors [9], and extend the discount range beyond 2 hops to infinity based on random walks [12]. However, these weighted one-mode projections still suffer from the structural limitations of the pairwise representation, which fails to preserve the edges holding multiple interactions of nodes in the bipartite network. More specifically, the pairwise one-mode projection loses information about physical connections shared by more than two nodes in the original bipartite network, as shown in the schematic of Fig 1. Note that in the muscle-centric network, the edges between muscles represent not only the relationships, but also the real bones, which should not be split into multiple parts or aggregated into a single one.

The interdependence associated with multiple connections in the body system can be represented by a hypergraph that includes higher order interactions [14]. Recently, several scientific activities have been devoted to hypergraph representations that capture multiple connections of the musculoskeletal system, where muscles are the nodes and bones are the hyper-edges [2, 15]. These approaches transfer the bipartite network of muscles and bones into a muscle-centric projection of the hypergraph, capturing more complex structural patterns in the body network than traditional pairwise models. However, previous studies seem to be restricted to limited hyper-edge representations, where the hyper-edges are transformed into a non-weighted binary network [15], or consider only the sizes of the hyper-edges, ignoring the degrees and hyper-degrees of the nodes in the networks [2]. Although many researchers have explored similar approaches for constructing a hypergraph from a bipartite graph in a more sophisticated way, such as the multiplex bipartite network embedding [16], to the best of our knowledge there are few applications to musculoskeletal networks.

An important question arising from previous studies is whether such a detailed higher-order representation can capture interwoven features in the connectivity of the human body system. Here, we give a positive answer to this question and show that hypergraphs can better distinguish typical muscles that bridge mesoscale structures in the human body compared to pairwise models. By incorporating node embedding methods developed for hypergraphs and

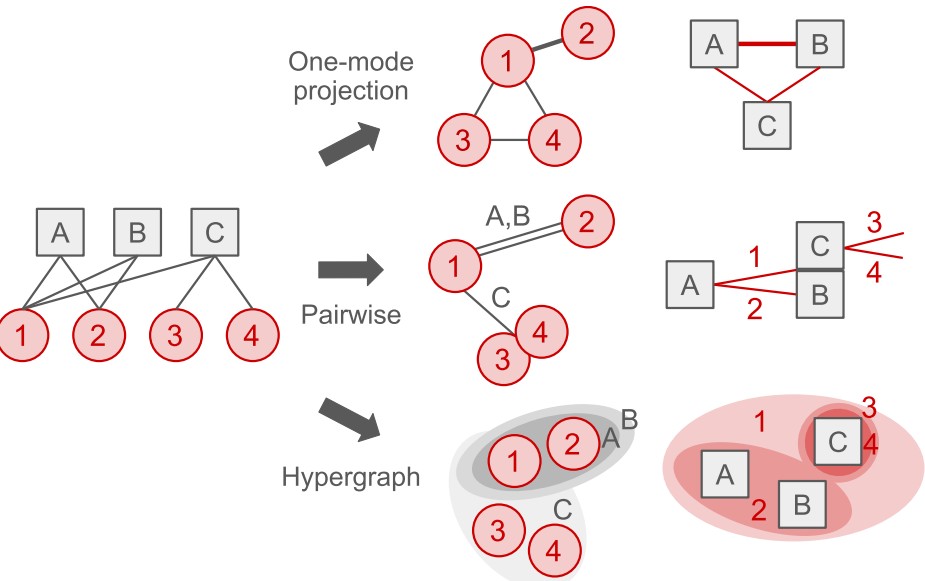

**Fig 1. Projections of bipartite network.** Bipartite graphs and their one-way projections into pairwise networks lose information about the complex interactions and structures of the human musculoskeletal system, where multiple bones connect to a muscle and vice versa.

pairwise networks, we investigate the ability of the network representation to capture neighborhood similarities. In this paper, we focus on 8 paradigmatic types of node embeddings that measure either proximity or structural properties of the networks described in the following section.

The contributions of this paper can be outlined as follows:

- Clarify the differences between pairwise networks and hypergraphs in terms of visualization, weighting schemes, and embedding methods.

- Demonstrate the superiority of hypergraph-based embedding methods in capturing complex muscle roles and connections across different parts of the body.

- Highlight practical advances in musculoskeletal studies, by demonstrating deeper insights into muscle functionality and interactions provided by hypergraph methods.

## Datasets

### Musculoskeletal network data

We used open-source data from the Network of Anatomical Texts (NAnaTex) [17], which integrates text-based anatomical information about bone and muscle interactions in the human body that is available in literature descriptions. The structural connections were defined based on the origins and insertions of the muscles [1]. Muscles with multiple origins/insertions were visually confirmed using a MeAV Anatomie 3D system (Panasonic Corporation and Okayama University) and included as distinct interactions, resulting in 1048 interactions of the 196 muscles. This covers more than 90% of the musculoskeletal interactions in the human body, and is available in the GitHub repository with the relevant files [17]. In this

**Table 1. Basic statics of musculoskeletal network datasets.**

| Data | Nodes | hyper-edges | $|V|$ | $|E|$ | $Avg.d(v)$ | $Avg.|e|$ | CCs | Larg.CC | GCC | Density | Diameter |
|------|-------|-------------|-------|-------|------------|-----------|-----|---------|-----|---------|----------|
| MbM | muscle | bone | 18 | 65 | 7.44 | 2.06 | 1 | 18 | 0.68 | 3.61 | 5 |
| BmB | bone | muscle | 65 | 18 | 2.06 | 7.44 | 1 | 65 | 0.89 | 0.28 | 6 |
| BB | bone | - | 85 | 105 | 2.47 | 2.00 | 6 | 64 | 0.20 | 0.03 | 30 |
| BM | bone, muscle | - | 85 | 143 | 3.36 | 2.00 | 1 | 85 | 0.00 | 0.04 | 12 |

Four datasets extracted from the musculoskeletal network visualized in Fig 3. MbM and BmB are the hypergraphs, and BB and BM are the pairwise networks built from the connectivity patterns of the body components. The network properties depend on how nodes and hyper-edges are defined and represented. Note that we only have muscle-bone and bone-bone connections in our body, not muscle-muscle. $|V|$ is the number of nodes. $|E|$ is the number of hyper-edges. $Avg.d(v)$ is the average degree of the nodes. $Avg.|e|$ is the average size of the hyper-edges (in pairwise networks, the size of the edges is always 2). CCs is the number of connected components. Larg.CC is the size of the largest connected component. GCC is the global clustering coefficient of all nodes in the clique expansion. Density is the ratio of hyper-edges to the number of nodes. Diameter is the length of the longest shortest path between nodes of the largest connected component in the hypergraph.

study, we focused on the basic musculoskeletal network of 18 muscles used in the previous study [15] and more detailed interactions with 65 bones in the NAnaTex.

Table 1 shows the basic statics of four datasets extracted from the musculoskeletal network visualized in Fig 3. Referring to the hypergraph structure analysis surveys [18, 19], we used nine measures for this study. For the representation of the musculoskeletal system, we examined two network models of the hypergraphs (MbM, BmB) and pairwise networks (BB, BM). As shown in the Table 3, pairwise networks had a lower global clustering coefficient and density than hypergraphs. In the following study, we used MbM to investigate the interdependencies between the muscles via the bones.

## Spanning muscles

To understand the structural interdependencies of the muscles, we focused on the muscles that bridge non-adjacent bones: the spanning muscles. Spanning muscles are located at the boundary of different parts of the body through the connections of the bones and play important roles in multiple body movements. In our dataset, we had two spanning muscles (deltoid, rectus femoris), as noted in bold in the Table 2. They are indicated by the blue colored muscles in Fig 2, which was obtained from the Visible Body computer software [20]. The deltoid is a thick, triangular muscle located in the human upper arm. It plays a crucial role in connecting the trunk to the arm, providing stability and movement to the shoulder. This muscle stabilizes the shoulder joint and allows movements such as raising the arm (flexion), spreading the arm to the side (abduction), and extending the arm backward (extension). The rectus femoris is one of the quadriceps muscles located on the front of the lower leg. It is the only one of the quadriceps that crosses both the trunk and the knee joint, helping to bring the leg closer to the body (flexion) and to straighten the knee (extension).

Spanning muscles such as the deltoid and rectus femoris play a pivotal role in facilitating a wide range of body movements and maintaining stability due to their position and connectivity in the musculoskeletal network. In the Table 2, we listed the 18 muscles used in this study with their abbreviations and the body parts they belong to, indicating the spanning muscles in bold.

## Materials and methods

### Summary of embeddings

Embedding methods aim to map nodes into a low-dimensional vector space while capturing the specific properties. The embedding criterion in networks can vary depending on the

**Table 2. Muscles connecting different body parts.**

| Muscles | Abb. | Body parts |
|---|---|---|
| Flexor Digitorum Superficialis | FDS | Forearm |
| Extensor Digitorum | ED | Forearm |
| **Deltoid** | **D** | Upper Arm |
| Biceps Brachii | BB | Upper Arm |
| Triceps Brachii | TRB | Upper Arm |
| Pectoralis Major | PMA | Trunk |
| Sternocleidomastoideus | SMA | Trunk |
| Longissimus | LO | Trunk |
| Latissimus Dorsi | LD | Trunk |
| Trapezius | TZ | Trunk |
| External Oblique | EO | Trunk |
| **Rectus Femoris** | **RF** | Upper Leg |
| Biceps Femoris | BF | Upper Leg |
| Vastus Lateralis | VL | Upper Leg |
| Adductor Longus | AL | Upper Leg |
| Tibialis Anterior | TA | Lower Leg |
| Gastrocnemius Medialis | GM | Lower Leg |
| Soleus | SOL | Lower Leg |

context and the practical application. It is primarily divided into two major types: proximity-based similarity (closeness between nodes) and structure-based similarity (topological connection patterns of nodes) [21]. Proximity-based similarity measures how close nodes are to each other based on their location in a network [22]. For instance, nodes connected by a short path or with many shared neighbors may be considered more similar. Deepwalk, node2vec, and HHE are the examples of the proximity-based method. On the other hand, structure-based similarity measures the similarity of nodes' structures based on their broader topological connection patterns within the network [21]. For example, when two nodes have comparable similarities in their degrees and the degrees of their respective neighbors, they can be considered similar, even if they are separated into different components of the network. struc2vec and HyperS2V are methods aligned with a structure-based criterion. We also employed three GNN-based approaches: GAT, HGNNP, and UniSAGE. Unlike other embedding methods, these GNN approaches require different settings for learning embeddings, which we will describe in detail at the end of this section.

We summarize the differences between the 8 embedding methods used in this study in Table 3.

## Deepwalk framework

Deepwalk [23] learns node embedding representations by treating random walks on a graph as sentences. It has two main steps:

1. Random Walk Generator: Generates random walks of a fixed length from each node, sampling the next node uniformly from the neighbors of the last node.

2. SkipGram Update: Uses the Skip-Gram language model [31] to maximize the probability of observing the context nodes in a window w around each node in the random walks. This model is based on the idea that words appearing in close proximity within a sequence are

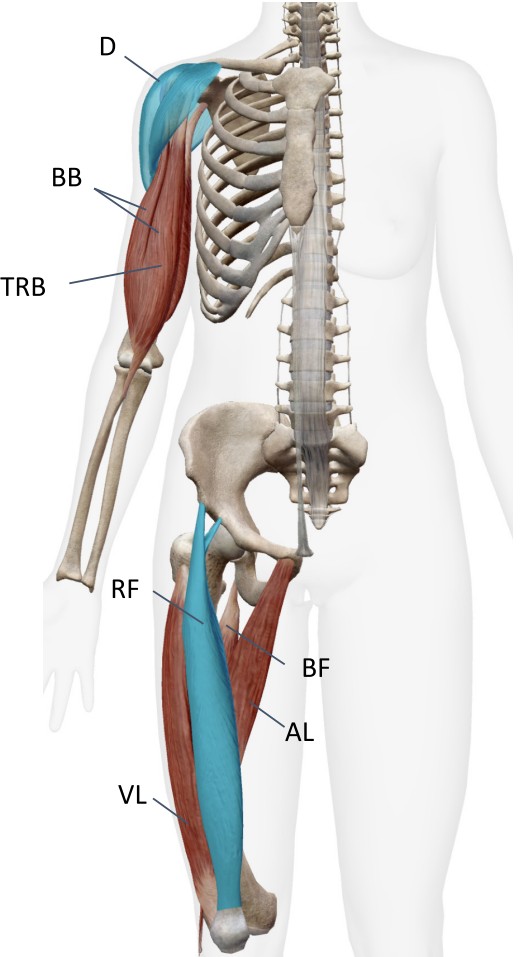

**Fig 2. Spanning muscles and bones.**

more likely to be related. By applying the Skip-Gram model to the random walk sequences, which consist of nodes near each other in the network, the learned embeddings capture rich proximity information. Hierarchical softmax is used to factorize the probabilities over a binary tree, allowing $O(\log|V|)$ computation. By leveraging the Skip-Gram model's ability

**Table 3. Summary of the embedding methods.**

| Method | Network type | Embedding criterion | Learning | Ref. |
| --- | --- | --- | --- | --- |
| Deepwalk | pairwise | proximity | unsupervised | [23] |
| node2vec | pairwise | proximity | unsupervised | [24] |
| GAT | pairwise | structure | supervised | [25] |
| struc2vec | pairwise | structure | unsupervised | [26] |
| HHE | hypergraph | proximity | unsupervised | [27] |
| HGNNP | hypergraph | proximity | supervised | [28] |
| UniSAGE | hypergraph | structure | supervised | [29] |
| HyperS2V | hypergraph | structure | unsupervised | [30] |

to capture contextual relationships, we can effectively encode the complex structural information in the multilayer graph into dense, low-dimensional embeddings.

In essence, Deepwalk learns node representations by encoding graph structures in a continuous vector space using techniques from language modeling, enabling powerful and scalable network analysis.

## node2vec framework

node2vec is a semi-supervised algorithm for learning continuous representations for nodes in networks. It builds on the Skip-gram model and extends the concept of using random walks to generate node sequences for learning embeddings.

Different from Deepwalk, node2vec introduces a biased random walk procedure that smoothly interpolates between two extreme sampling strategies (Breadth-first Sampling and Depth-first Sampling). It generates fixed-length random walks starting from each node. The walk is biased, using parameters to balance the exploration-exploitation tradeoff.

Same as Deepwalk, node2vec uses the Skip-gram model to learn embeddings from the generated random walks.

## struc2vec framework

struc2vec [26] is a method for simple networks that evaluates the structural characteristics of nodes based on their degrees. For nodes $u$ and $v$, it calculates distances considering:

Self-node (0-hop) distance:

$$D^0(u, v) = \frac{\max(d_u, d_v)}{\min(d_u, d_v)} - 1 \tag{1}$$

k-hop distances using a recurrence relation:

$$D^k(u, v) = D^{k-1}(u, v) + DTW(D^0, SD^k(u), SD^k(v)), k > 0 \tag{2}$$

struc2vec constructs a multilayer weighted graph where each layer has $|V|$ nodes. Intra-layer edge weights are

$$w_k(u, v) = e^{-D^k(u,v)}. \tag{3}$$

Inter-layer connections exist between the same nodes across layers.

Random walks are performed on this multilayer graph with defined transition probabilities for both intra-layer and inter-layer movements. The generated context sequences are used with the Skip-Gram model to create node embeddings, providing a structural representation of the nodes for downstream analysis and modeling tasks.

## HyperS2V framework

HyperS2V [30] is an embedding method for hypergraphs that preserves structural information. Key concepts include:

1. Hyper-degree (HD): Represents node degree information in a hypergraph.

$$\text{HD}_i = \text{sort}(\{\sum_v I_{ve} | \forall e, I_{ie} = 1\}), \tag{4}$$

where $I$ denotes the incidence matrix, and $\sum_v I_{ve}$ represents the size of edge $e$, and sort($\cdot$) is a function that sorts the elements in descending order.

2. Magnitude-position distance (MPD): Assesses distance between two hyper-degrees.

$$\text{MPD}(s_{u_i}, s_{v_j}) = \exp\left(\sqrt{\left(1 - \frac{\min(s_{u_i}, s_{v_j})}{\max(s_{u_i}, s_{v_j})}\right)^2 + |b_{u_i} - b_{v_j}|^2}\right) - 1 \tag{5}$$

$s_{u_i}$ and $s_{v_j}$ are the $i$th and $j$th elements in $\text{HD}_u$ and $\text{HD}_v$, respectively, representing the sizes of the $i$th and $j$th largest edges connected to $u$ and $v$. $b_{u_i}$ and $b_{v_j}$ are bias terms indicating the "positional importance" of $s_{u_i}$ and $s_{v_j}$ in the descending-order sorted lists $\text{HD}_u$ and $\text{HD}_v$, calculated as:

$$b_s = \frac{1}{\max(\text{HD}) - s + 1} \tag{6}$$

3. 0-hop distance: Uses dynamic time warping (DTW) with MPD to compare HDs.

$$D^0(u, v) = \text{DTW}(\text{MPD}; \ \text{HD}_u, \ \text{HD}_v). \tag{7}$$

4. k-hop distance: Considers neighboring nodes' connection patterns.

$$D^k(u, v) = D^{k-1}(u, v) + \text{DTW}(D^0; \ \text{NCHD}^k(u), \ \text{NCHD}^k(v)). \tag{8}$$

HyperS2V constructs a multilayer weighted graph using these distances, performs random walks to generate context sequences, and learns node embeddings using the Skip-Gram model.

This method differs from struc2vec in how it captures neighborhood similarity in hypergraphs.

## HHE framework

HHE (Hypergraph-based Heterogeneous Embedding) [27] is specifically developed to address document recommendation tasks where complex interactions between heterogeneous entities are represented as hyperedges. The core of the HHE algorithm lies in its eigendecomposition approach. Specifically, it seeks the first $k$ generalized eigenvectors that correspond to the $k$ smallest nonzero eigenvalues of a normalized Laplacian matrix of the hypergraph, which encapsulates the structural properties of the hypergraph in a mathematical form. This approach ensures that the resulting embeddings preserve the essential topological features of the original hypergraph in a lower-dimensional space.

In the context of document recommendation, HHE's ability to capture complex relationships between different types of entities (such as documents, authors, keywords, and categories) in a unified embedding space has shown promising results. This makes it a valuable tool for researchers and practitioners working with heterogeneous data in recommendation systems and other related fields.

## GNNs for embedding generation

As the end-to-end frameworks, the Graph Neural Network (GAT) and Hypergraph Neural Network (HGNNP and UniSAGE) frameworks require node features, target labels, and a

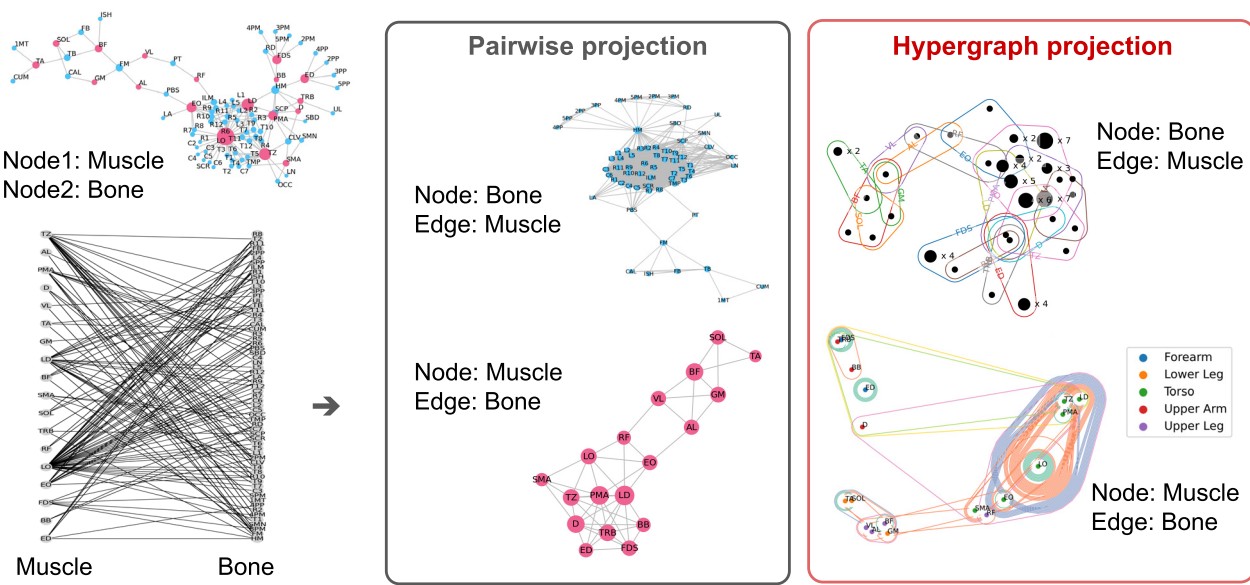

**Fig 3. musculoskeletal network (18 muscles and 65 bones).**

prediction task to learn embeddings. For a fair comparison with other embedding methods, we used the adjacency matrix as node features and node degrees as target labels. These methods learn embedding representations through a regression task to predict node degrees. Specifically, we added a fully connected readout layer to the end of these models to predict node degrees. We use the hidden features as embedding outputs. The model structures and hyperparameters were kept at their default settings for each method.

## Results

### Six network representations of the musculoskeletal system

Fig 3 shows the visualizations of the six types of network representations of 18 muscles and 65 bones from one side of the human body, corresponding to the datasets in Table 1. The upper left figure shows the pairwise network of heterogeneous nodes of muscles and bones, colored in pink and right blue, respectively (BM dataset). The lower left figure shows the bipartite network of muscles and bones. The middle two figures are pairwise networks and the right two figures are hypergraphs, both projected from the lower left bipartite network, respectively. The top is the bone-centric projections (BmB dataset), and the bottom figure shows the muscle-centric projections (MbM dataset).

We visualized the hypergraph representations in two ways. In the top figure, the nodes represented as black dots with identical hyper-edge membership are collapsed into a single point whose size is increased to reflect the number of members [32]. The hyper-edges are enclosing circles containing the nodes that belong to the edge set. In the lower figure, the node sizes are the same but colored by the corresponding body parts. The position of the nodes in the lower figure is derived from the 2-dimensional embeddings of HyperS2V. Thus, the distances between node pairs indicate their structural similarity (i.e., the closer two nodes are, the more structurally similar they are.) An enlarged version of this figure is included in the S3 Fig along with different embedding methods from struc2vec S4 Fig, which is the preceding method of

HyperS2V. In the following study, we focused on the role of muscles and used the MbM dataset for analysis.

## Hypergraph representation

The fundamental difference between a hyper-edge in a hypegraph and an ordinary edge in a pairwise network is that the former has three properties: size, degree, and hyper-degree, where size is the number of nodes contained in a hyper-edge, degree is the number of hyper-edges in which a node is contained, and hyper-degree is a sorted list representing the sizes of each hyper-edge in which a node is contained. For example, in the simple musculoskeletal network of 18 muscles and 65 bones, the deltoid, the spanning muscle at the shoulder girdle that connects the upper arm to the trunk, is contained within three hyper-edges of bones that connect a varying number of muscles. Therefore, the 0-hop hyper-degree (i.e., muscles in the same hyper-edges) of the deltoid is [6, 5, 4], which is a sorted list of the sizes of each of the three hyper-edges connecting the deltoid to other muscles.

Fig 4 shows the adjacency matrices and hyper-degrees of the muscle-centric hypergraph with 18 muscles as nodes and 65 bones as hyper-edges obtained from the MbM dataset. The left and middle adjacency matrices are colored with different weighting schemes, indicated by the thickness of the blue color. In the left figure, the weights are the number of neighbor muscles of the muscles (i.e., the degrees), and there are no weights on the diagonal of the matrix. In the middle figure, the weights are the common neighbor bones of the muscle pairs (i.e., the intersection size of overlapping hyper-edge pairs) or the number of bones connected to the muscle (i.e., the size of the hyper-edges), which is displayed on the diagonal of the matrix. The right figure shows the hyper-degrees of the same muscle-centric hypergraph. The x-axis shows the parameter $s$, which restricts the size of the edges to be considered, and the weights show the number of edges with size $\geq s$. In the MbM dataset, there were no muscles with more than size $\geq 7$ hyper-edges.

## Comparing different embeddings of the muscles

We applied 8 embedding methods of Deepwalk, node2vec, GAT, struc2vec, HHE, HGNNP, UniSAGE, and HyperS2V to the MbM dataset. Since Deepwalk, node2vec, GAT, and struc2-vec are methods for pairwise networks, we fed the clique expansion network of the MbM

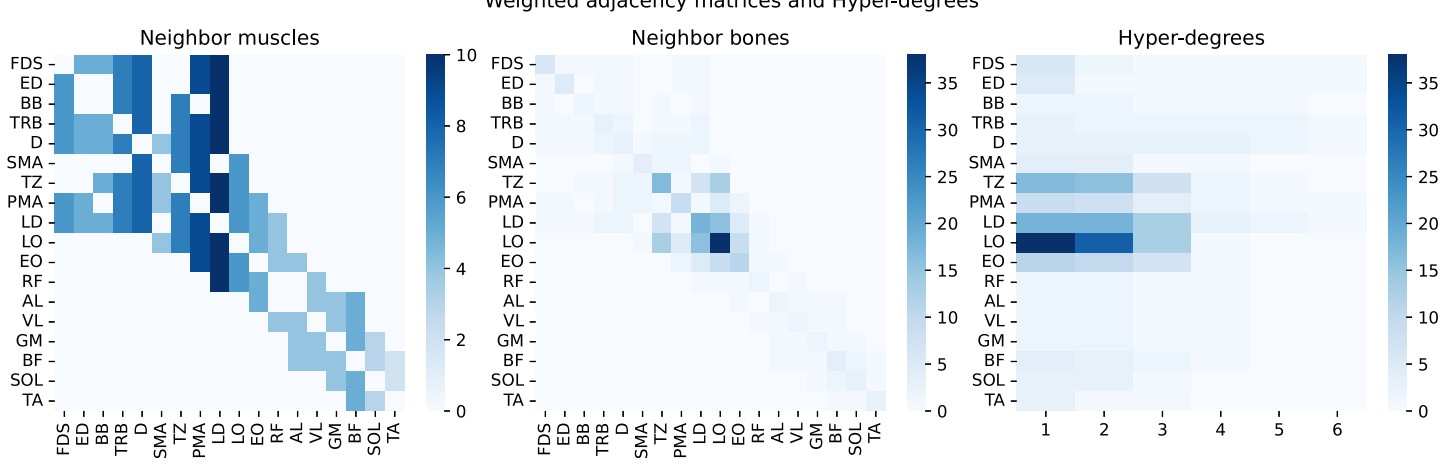

**Fig 4. Weighted adjacency matrices and hyper-degrees of the 18 muscles.**

dataset as the input network. We standardized the 16-dimensional embedding vectors in each method and computed the Euclidean distance to determine how similar two muscles are in each embedding result. We chose Euclidean distance to consider the nature of representation generation in these selected methods (i.e., similar nodes are placed close to each other in the vector space, and vice versa).

Specifically, we calculated the distance between the target muscles (upper arm, upper leg), and the other muscles listed on the x-axis including the target muscles, as shown in Figs 5 and 6. For upper arm muscles, HyperS2V showed a difference between deltoid (D), biceps femoris (BF), and triceps brachii (TRB), while the other embedding methods tended to characterize them together. For upper leg muscles, Deepwalk, node2vec, struc2vec, and HyperS2V discriminated the rectus femoris (RF) from the other muscles, while GAT, HHE, HGNNP, and Uni-SAGE showed the almost no difference between the four target muscles. We noticed that HHE showed high distances in almost every muscle pair, indicating that there was little correlation between the embeddings of the muscles. On the contrary, the GNN-based methods of GAT, HGNNP and UniSAGE did not show much difference as their distances in each muscle pair were close to zero.

These results are consistent with the fact that both D and RF are the spanning muscles that connect different parts of the body and support multiple body movements, as we discussed earlier. In addition, we found that only in HyperS2V did the sternocleidomastoideus (SMA) have the smallest distance (i.e., most similar) to the D and RF, consistent with the fact that the SMA is the muscle that connects the trunk to the head and supports both flexion and extension of the neck. We also noticed that the longissimus (LO) showed comparatively small distances in all three methods. This tendency is consistent with the fact that the LO has the most extensive connections to the bones in our datasets (right figure in Fig 4), extending along the spine and connecting directly to many vertebrae, ribs, and the pelvis.

## Discussion

The complex interconnectivity of the human musculoskeletal system can be represented by different network models, depending on the definition of the nodes and edges of interest. In this study, we focused on the role of muscles and used the MbM dataset, where the nodes are the 18 muscles, and the edges are the 65 bones. By examining six network representations, we clarified that the hypergraphs can preserve information about many-to-many connections of the musculoskeletal system that cannot be represented by pairwise models. To compare the representability of the network models, we used 8 embedding methods designed for both pairwise and hypergraph models, showing the superior performance of the hypergraph-based HyperS2V. There are basically two methods for transforming hypergraphs into pairwise networks; star and clique expansions and their generalization [33]. In this study, we used the clique expansion technique, which is equivalent to the muscle-centric one-mode projection of the bipartite network of muscles and bones. Since the clique expansion forms a complete graph connecting all nodes within the same hyperedge, it is equivalent to the one-mode projection connecting all nodes that have a common neighbor, i.e., a hyperedge. We chose the clique expansion (one-mode projection) over the star expansion because it is more suitable for applications that focus on the relationships between muscles. In star expansion, a new central node is introduced for each hyperedge that connects to the associated muscles, which is equivalent to reverting to the original bipartite network of muscles and bones. This can obscure the direct relationships between muscles by shifting some of the focus to the central nodes (bones), thus detracting from a muscle-centric analysis.

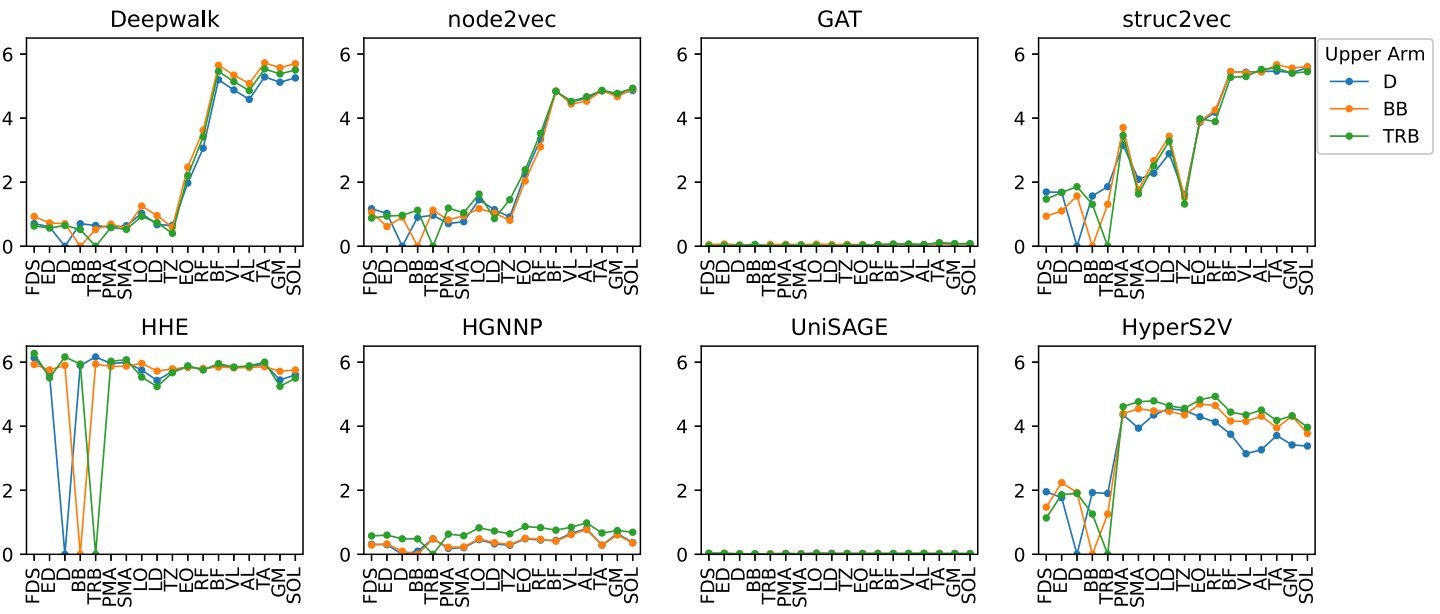

**Fig 5. Distance with the upper arm muscles.**

The experimental result that HyperS2V discriminated the differences between the upper arm muscles in the similarity of embeddings with the trunk muscles suggests its ability to capture more nuanced musculoskeletal connectivity. In fact, the upper arm muscles of D, BF, and TRB are all directly or indirectly connected to the trunk muscles, but only D, located at the upper arm-trunk interface, supports complex movements of the shoulder joint. It is, therefore, more difficult to distinguish the role of D in the upper arm muscles. On the other hand, RF is the only upper leg muscle that acts on both the knee and the hip joints, directly connecting from the trunk to the upper leg [34], and all three embedding methods identified its difference. The finding that only HyperS2V distinguished the similarity of SMA to the spanning muscles of D and RF is also interesting because SMA is also the muscle that connects different body parts of the trunk and head. These results support the idea that HyperS2V is able to capture the similarities of the complex structural roles of distant muscles in the body system, which most proximity or GNN-based embedding methods fail to capture. For GAT, HGNNP, and UniSAGE, the distances between muscles have converged to nearly zero. This is due to either an extremely polarized distance distribution or the distances between all node pairs being close to zero. We attribute this phenomenon to the inherent nature of these GNN frameworks, where over-smoothing occurs as a result of the task design for generating embeddings from the network structure. As shown in S1 and S2 Figs, the shapes of the line graph of the embedding vectors were almost the same in GAT, almost the same but different in magnitude of the values in HGNNP and UniSAGE. Conversely, in HHE there was almost no correlation in the shape of the vectors for each muscle, resulting in the high distances shown in Figs 5 and 6.

For network analysis of living systems, hypergraph is a new direction that helps to more accurately model the interacting networks of higher-order topology represented at different levels in living systems [35]. Recent studies of human musculoskeletal networks have revealed the relationship between anatomical and functional properties in the coordination of human body systems, using a hypergraph representation of the anatomical network. [2, 15]. However, they applied one-mode projection, i.e., clique expansion to analyze the structural property of

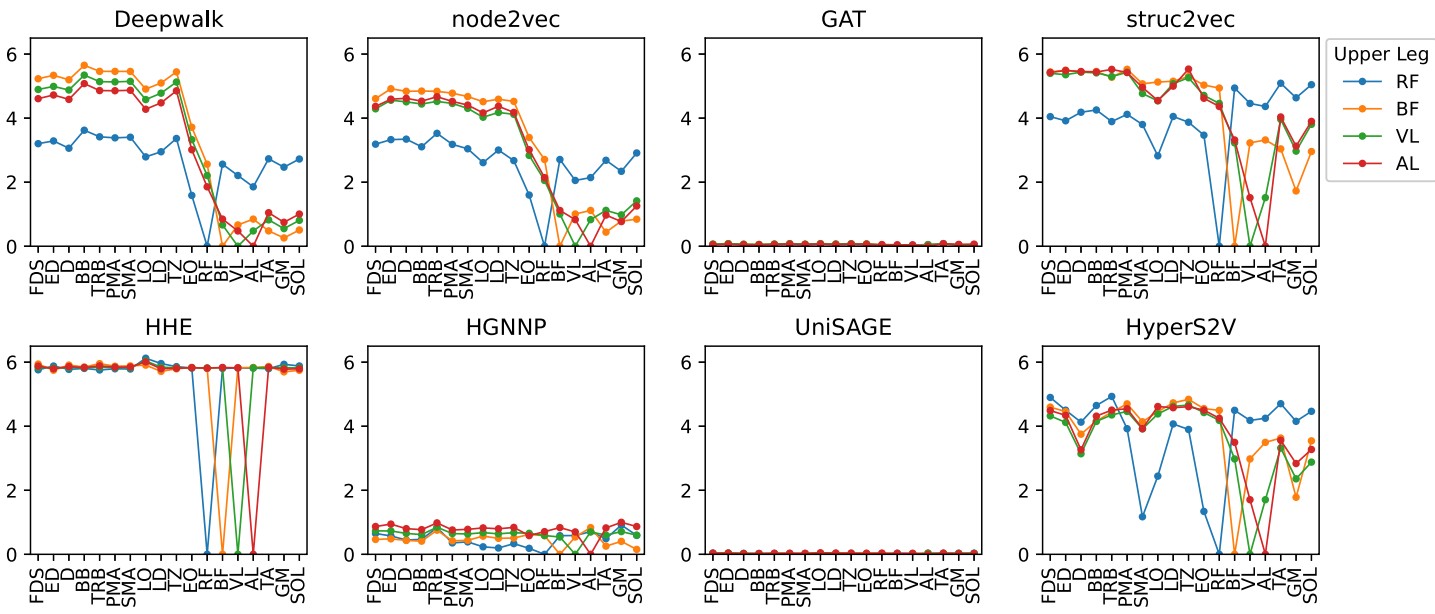

**Fig 6. Distance with the upper leg muscles.**

the network, which cannot cover the higher-order substructures such as hyperedges [29], nor did they used embedding methods to explore the differences in the expressiveness of the network models. In addition, these previous studies have focused on theoretical predictions or cluster-level analysis, which is separate from the specific structural role of each muscle in the interconnectivity of the body system found in this study.

The limitation of this study also lies in the hypergraph approach to investigate the higher-order topology of the whole body network; a complex system of heterogeneous components that interact with each other with different types of relationships. In our analysis, we focused on the connectivity between muscles and bones, ignoring the connections between bones. While most muscles are not directly connected, bones are connected to each other and play an essential role in supportting the body systems, which cannot be included in hypergraph models because they are projected from bipartite networks of muscles and bones. We used only a limited part of the anatomical network data in this study and will include more detailed and multiple types of datasets that can be represented by multilayer network models [36]. As we have shown in the S1 Table, NAnaTex has more and larger datasets that reflect the complex interconnectivity of the body, which we plan to use for the future study.

Hypergraphs and multilayer network models might be used in a complementary way to understand the indirect or distant relationships between multiple components in living systems, a recurring problem in clinical and rehabilitation settings. For example, the secondary compensatory effects of an injury on other regions that are not necessarily adjacent to each other in the anatomical network [2, 37, 38]. One way to understand complex neuromuscular control [5, 39], including secondary compensations, is to combine anatomical networks of physical connections between muscles with functional networks based on the synchrony of muscle potentials calculated from electromyogram (EMG) signals [15, 40]. In future research, we will integrate these functional network data to validate advanced network models and explore their applications in understanding the complex interdependencies within living systems. This approach aims to improve our understanding of the intricate relationships beyond

individual components, ultimately contributing to a more holistic understanding of the neuro-musculoskeletal system.

## Conclusion

In this study, we addressed the problem of network representation and its learning by the embedding method of the human musculoskeletal system. We applied HyperS2V, a hypergraph-based embedding method, to elucidate the positional importance of muscles spanning different parts of the body. Experiments demonstrated the superiority of our method over the pairwise-based embedding methods of Deepwalk and struc2vec. Despite their importance, few studies have addressed the positional importance of muscles in the whole-body structure of interwoven musculoskeletal networks. We believe this work provides new opportunities for the community.

## Supporting information

**S1 Fig. Embedding vector values of upper arm muscles.** (16 dimensions). (PNG)

**S2 Fig. Embedding vector values of upper leg muscles.** (16 dimensions). (PNG)

**S3 Fig. Musculoskeletal hypergraph with nodes displayed by HyperS2V.** (PDF)

**S4 Fig. Musculoskeletal hypergraph with nodes displayed by struc2vec.** (PDF)

**S1 Table. Basic statics of musculoskeletal network datasets (full-data).** (PNG)

## Acknowledgments

We thank professor Ryusuke Momota for providing musculoskeletal network data of NAna-Tex [17] and his helpful comments on this study. The content is solely the responsibility of the authors and does not necessarily represent the official views of any of the funding agencies.

## Author Contributions

**Conceptualization:** Hiroko Yamano.

**Data curation:** Hiroko Yamano.

**Formal analysis:** Hiroko Yamano.

**Investigation:** Hiroko Yamano, Shu Liu.

**Methodology:** Shu Liu.

**Project administration:** Hiroko Yamano.

**Supervision:** Fujio Toriumi.

**Validation:** Hiroko Yamano.

**Visualization:** Hiroko Yamano, Shu Liu.

**Writing – original draft:** Hiroko Yamano, Shu Liu.

**Writing – review & editing:** Hiroko Yamano, Shu Liu, Fujio Toriumi.

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
