## [Decision Letter · Decision Letter 0]

1 Jul 2024

PONE-D-24-20578Hypergraph modeling of complex interactions: Applications from human musculoskeletal structures to complex system dynamicsPLOS ONE

Dear Dr. yamano,

Thank you for submitting your manuscript to PLOS ONE. After careful consideration, we feel that it has merit but does not fully meet PLOS ONE’s publication criteria as it currently stands. Therefore, we invite you to submit a revised version of the manuscript that addresses the points raised during the review process.

We look forward to receiving your revised manuscript.

Kind regards,

Hocine Cherifi

Academic Editor

PLOS ONE

Journal Requirements:

3. We notice that your supplementary tables are included in the manuscript file. Please remove them and upload them with the file type 'Supporting Information'. Please ensure that each Supporting Information file has a legend listed in the manuscript after the references list.

Reviewers' comments:

Reviewer's Responses to Questions

**Comments to the Author**

1. Is the manuscript technically sound, and do the data support the conclusions?

Reviewer #1: Yes

Reviewer #2: Partly

2. Has the statistical analysis been performed appropriately and rigorously? 

Reviewer #1: Yes

Reviewer #2: N/A

3. Have the authors made all data underlying the findings in their manuscript fully available?

Reviewer #1: Yes

Reviewer #2: No

4. Is the manuscript presented in an intelligible fashion and written in standard English?

Reviewer #1: Yes

Reviewer #2: Yes

5. Review Comments to the Author

**Reviewer #1: **This paper focuses on utilizing graphs and hypergraphs to model the complex system of human musculoskeletal structures. Additionally, it compares the performance of simple graph and hypergraph-based modeling methods using various existing representation learning techniques. The experimental results demonstrate that hypergraph modeling outperforms simple graph modeling. Overall, this is a well-conceived application and exploration paper that investigates the use of graph embeddings in the biomedical domain.

I have a few comments to help the authors improve their paper:

1. It would be beneficial to employ more simple graph embedding and hypergraph embedding models (not just three) on the constructed graphs and benchmark their performance.

2. The method of constructing a hypergraph from a bipartite graph is similar to that described in the paper "Multiplex Bipartite Network Embedding using Dual Hypergraph Convolutional Networks."

3. There are two methods for transforming between simple graphs and hypergraphs, star and clique expansion. The authors should mention these methods and consider employing UniGNN on their datasets and tasks. See "UniGNN: a Unified Framework for Graph and Hypergraph Neural Networks."

4. Another relevant paper proposes a Line Expansion from hypergraphs. The authors could consider adding this information to their paper: "Semi-supervised Hypergraph Node Classification on Hypergraph Line Expansion."

**Reviewer #2: **This paper models the complex many-to-many interactions between muscles and bones as a hypergraph. The embeddings of the nodes are then learned using two graph-based and one hypergraph-based unsupervised embedding methods. The authors analyze the results based on the Euclidean distances between the obtained embeddings.

While the paper presents an interesting approach to modeling human musculoskeletal structures as hypergraphs, there is room for improvement.

Strenghts.

- The rationale for modeling the musculoskeletal structure as hypergraphs is well motivated.

- The authors clearly explain how their appraoch is distinct from previous approaches.

- The figures effectively describe the concepts discussed.

Weaknesses and Questions.

- In the introduction, the authors mention one-mode projection. Is this equivalent to clique expansion?

- The datasets used in the paper are too small. Please consider using larger datasets if possible.

- Only a small number of embedding methods are employed. Other graph-based methods such as node2vec and hypergraph-based methods such as hyper2vec or VilLain could also be considered. Are the results consistent when using these additional methods? Please refer to the references below.

- The embedding methods discussed can also be applied in star expansion. How does that impact the results?

- While the analysis of the results using the embedding methods is the main contribution of the paper, the discussion of the methods themselves is overly detailed. I recommend simplifying the description of the methods and focusing more on the experimental resutls and analyses.

- For better readability, please structure the discussion more clearly.

I recommend the authors revise the manuscript to enhance clarity and focus.

References.

- node2vec: Scalable feature learning for networks (KDD 2016)

- Hyper2vec: Biased random walk for hyper-network embedding (DSAFAA workshop 2019)

- VilLain: Self-Supervised Learning on Homogeneous Hypergraphs without Features via Virtual Label Propagation (WWW 2024)

6. PLOS authors have the option to publish the peer review history of their article (what does this mean?). If published, this will include your full peer review and any attached files.

Reviewer #1: No

Reviewer #2: No

---

## [Author Response · Author response to Decision Letter 0]

17 Aug 2024

Reviewer #1: This paper focuses on utilizing graphs and hypergraphs to model the complex system of human musculoskeletal structures. Additionally, it compares the performance of simple graph and hypergraph-based modeling methods using various existing representation learning techniques. The experimental results demonstrate that hypergraph modeling outperforms simple graph modeling. Overall, this is a well-conceived application and exploration paper that investigates the use of graph embeddings in the biomedical domain.

I have a few comments to help the authors improve their paper:

1. It would be beneficial to employ more simple graph embedding and hypergraph embedding models (not just three) on the constructed graphs and benchmark their performance.

>> Following your suggestion, we employed five additional embedding methods to compare their performance. The main differences between the 8 methods, such as network type (pairwise / hypergraph), embedding (proximity / structure), and learning (supervised / unsupervised) criteria, are shown in Table 3. 

2. The method of constructing a hypergraph from a bipartite graph is similar to that described in the paper "Multiplex Bipartite Network Embedding using Dual Hypergraph Convolutional Networks."

>> Thank you for your informative proposal. We have reviewed the proposed article and agree with the reviewer that we have a similar understanding with the authors about transforming a bipartite network into a hypergraph. We have included this in the introduction, citing the article as follows:

"Although some researchers have employed similar approaches to construct a hypergraph from a bipartite graph in a more sophisticated way, such as the multiplex bipartite network embedding \\cite{xue2021multiplex}, to the best of our knowledge there are few applications to musculoskeletal networks." 

3. There are two methods for transforming between simple graphs and hypergraphs, star and clique expansion. The authors should mention these methods and consider employing UniGNN on their datasets and tasks. See "UniGNN: a Unified Framework for Graph and Hypergraph Neural Networks."

>> Thank you for your insightful suggestion. We explained why we chose the clique expansion for pairwise based applications and described the correspondence between the star expansion and the bipartite network as follows. We also learned how UniGNN integrates the information of the hyperedges without using clique expansion, and employed UniSAGE, which is one of the methods in UniGNN. We chose UniSAGE because it is the default method in the four embedding methods collectively referred to as UniGNN. 

"In this study, we used the clique expansion technique, which is equivalent to the muscle-centric one-mode projection of the bipartite network of muscles and bones. Since the clique expansion forms a complete graph connecting all nodes within the same hyperedge, it is equivalent to the one-mode projection connecting all nodes that have a common neighbor, i.e., a hyperedge. We chose the clique expansion (one-mode projection) over the star expansion because it is more suitable for applications that focus on the relationships between muscles. In star expansion, a new central node is introduced for each hyperedge that connects to the associated muscles, which is equivalent to reverting to the original bipartite network of muscles and bones. This can obscure the direct relationships between muscles by shifting some of the focus to the central nodes (bones), thus detracting from a muscle-centric analysis."

4. Another relevant paper proposes a Line Expansion from hypergraphs. The authors could consider adding this information to their paper: "Semi-supervised Hypergraph Node Classification on Hypergraph Line Expansion."

>> We surveyed the recommended paper and interested in the Line Expansion method, which concerns the information loss in hypergraph transformation, overcoming the limitation of clique and star expansion. However, in the biological settings of musculoskeletal network in this study, the nodes and edges should not be divided into multiple parts, because the nodes and edges in our study is not only the relationships, but also the physical connections of real muscles and bones. Therefore, for the same reason with clique and star expansions, we think that Line Expansion also loses the hypergraph nature by its expansion. However, given the importance of clarifying the expansion methods, we have cited the recommended paper in the sentence below.

"There are basically two methods for transforming hypergraphs into pairwise networks; star and clique expansions and their generalization \\cite{yang2022semi}. "

=======

Reviewer #2: This paper models the complex many-to-many interactions between muscles and bones as a hypergraph. The embeddings of the nodes are then learned using two graph-based and one hypergraph-based unsupervised embedding methods. The authors analyze the results based on the Euclidean distances between the obtained embeddings.

While the paper presents an interesting approach to modeling human musculoskeletal structures as hypergraphs, there is room for improvement.

Strenghts.

- The rationale for modeling the musculoskeletal structure as hypergraphs is well motivated.

- The authors clearly explain how their appraoch is distinct from previous approaches.

- The figures effectively describe the concepts discussed.

Weaknesses and Questions.

- In the introduction, the authors mention one-mode projection. Is this equivalent to clique expansion?

>> Thank you for the clarification. We have added the response as the following sentences in the discussion.

"In this study, we used the clique expansion technique, which is equivalent to the muscle-centric one-mode projection of the bipartite network of muscles and bones. Since the clique expansion forms a complete graph connecting all nodes within the same hyperedge, it is equivalent to the one-mode projection connecting all nodes that have a common neighbor, i.e., a hyperedge."

- The datasets used in the paper are too small. Please consider using larger datasets if possible.

>> This study aims to compare the representation performance of different embedding methods on small datasets of major muscles and bones. Therefore, we believe that the application of larger datasets is beyond the scope of this study. In the future analysis, we plan to use larger datasets, which are described in the revised sentences below: 

"We used only a limited part of the anatomical network data in this study and will include more detailed and multiple types of datasets that can be represented by multilayer network models\\cite{de2023more}. As we have shown in the Supporting information \\nameref{S1_Table}, NAnaTex has more and larger datasets that reflect the complex interconnectivity of the body, which we plan to use for the future study."

- Only a small number of embedding methods are employed. Other graph-based methods such as node2vec and hypergraph-based methods such as hyper2vec or VilLain could also be considered. Are the results consistent when using these additional methods? Please refer to the references below.

References.

- node2vec: Scalable feature learning for networks (KDD 2016)

- Hyper2vec: Biased random walk for hyper-network embedding (DSAFAA workshop 2019)

- VilLain: Self-Supervised Learning on Homogeneous Hypergraphs without Features via Virtual Label Propagation (WWW 2024)

>> Thank you for the valuable information. Following your suggestion, we employed five additional embedding methods, including node2vec, to compare their performance. The main differences between the 8 methods, such as network type (pairwise / hypergraph), embedding (proximity / structure), and learning (supervised / unsupervised) criteria, are shown in Table 3. Due to time constraints,　we added the embedding methods we were familiar with in this revision. We employed HGNNP and UniSAGE, instead of hyper2vec and VilLain, which are also hypergraph-based, but learn proximity and structural features, respectively. 

- The embedding methods discussed can also be applied in star expansion. How does that impact the results?

>> Thank you for your insightful comment. We have added the response as the following sentences in the discussion.

"We chose the clique expansion (one-mode projection) over the star expansion because it is more suitable for applications that focus on the relationships between muscles. In star expansion, a new central node is introduced for each hyperedge that connects to the associated muscles, which is equivalent to reverting to the original bipartite network of muscles and bones. This can obscure the direct relationships between muscles by shifting some of the focus to the central nodes (bones), thus detracting from a muscle-centric analysis."

- While the analysis of the results using the embedding methods is the main contribution of the paper, the discussion of the methods themselves is overly detailed. I recommend simplifying the description of the methods and focusing more on the experimental resutls and analyses.

>> Following the reviewer's recommendation, we have revised the method to describe it more simply.

- For better readability, please structure the discussion more clearly.

I recommend the authors revise the manuscript to enhance clarity and focus.

>> Thank you for the useful suggestion. According to the reviewer's suggestion, we have carefully read the entire structure of the discussion and revised the sentences.

---

## [Editor Report · Decision Letter 1]

27 Aug 2024

Hypergraph modeling of complex interactions: Applications from human musculoskeletal structures to complex system dynamics

PONE-D-24-20578R1

Dear Dr. yamano,

We’re pleased to inform you that your manuscript has been judged scientifically suitable for publication and will be formally accepted for publication once it meets all outstanding technical requirements.

Kind regards,

Hocine Cherifi

Academic Editor

PLOS ONE
---

## [Editor Report · Acceptance letter]

10 Sep 2024

PONE-D-24-20578R1 

PLOS ONE

Dear Dr. Yamano, 

I'm pleased to inform you that your manuscript has been deemed suitable for publication in PLOS ONE. Congratulations! Your manuscript is now being handed over to our production team.

Kind regards, 

on behalf of

Professor Hocine Cherifi 

Academic Editor

PLOS ONE